# Does Involution Cause Anxiety? An Empirical Study from Chinese Universities

**DOI:** 10.3390/ijerph19169826

**Published:** 2022-08-09

**Authors:** Da Yi, Jingwen Wu, Minqiang Zhang, Qing Zeng, Jinqing Wang, Jingdan Liang, Yashi Cai

**Affiliations:** 1School of Psychology, South China Normal University, Guangzhou 510631, China; 2Key Laboratory of Brain, Cognition and Education Sciences, Ministry of Education, South China Normal University, Guangzhou 510631, China; 3School of Psychology, Center for Studies of Psychological Application, South China Normal University, Guangzhou 510631, China; 4Guangdong Key Laboratory of Mental Health and Cognitive Science, South China Normal University, Guangzhou 510631, China

**Keywords:** college students, involution behavior scale, active involution, passive involution, anxiety

## Abstract

The debate over whether involution causes anxiety has persisted because no studies have attempted to quantify introversion and study its relationship to anxiety. This study quantified involution and explored its relationship with anxiety, provided evidence about whether involution was related to anxiety, and created a foundation for other scholars to carry out research on involution. Interviews and questionnaires were conducted to investigate the characteristics of 535 Chinese college students’ involution behavior and its relationship with anxiety. We found that involution was not necessarily positively related to anxiety. The specific results were as follows: (1) The involution behavior of the Chinese college students could be divided into three types: the passive involution, reward-oriented involution, and achievement-motivated involution; (2) Significant differences in the involvement of involution existed at the college level; (3) Three motivations that resulted in involution, from primary to secondary, were achievement-motivation, reward-orientation, and passive engagement; and (4) Passive involution, reward-oriented involution, and the total scores for the involution behavior of the college students were significantly and positively correlated with anxiety. Among the three types of involution behavior, the college students’ passive involution had a significant and positive predictive effect on their anxiety, while achievement-motivated involution had a significant and negative predictive effect.

## 1. Introduction

The term “neijuan” (in English, “involution”) has become increasing widely used among new generations of Chinese people to describe their current high-pressure academic and work conditions. A search for “neijuan” on Chinese Bing yielded more than 100 million references and “neijuan” was ranked one of the top 10 Chinese buzzwords of 2020 [1]. Academically, involution was originally defined as a state of stagnation that is encountered during the development of a social or cultural pattern, or the failure to promote [2]. However, with the widespread usage of “neijuan”, or involution, people are now accustomed to regarding it as an irrational and inefficient competition for high-quality resources. 

The phenomenon of involution was first realized among college students in their irrational academic competition for better scores. Then, it expanded to other areas of life, including education, work, entertainment, and even marriage. A general consensus is that involution is highly related to anxiety, based on more than 100 million quotations on the internet pertaining to both involution and anxiety. Several studies on the relationship between involution and anxiety have been published in the field of sociology [2,3]. However, most of the relevant discussions have been based on social review articles, which lack the powerful support of specific data analyses, and their definitions of involution remain in conformity with the definition of “neijuan” in the network context. 

The involvement of Chinese college students in applying the concept of involution to their symptoms of anxiety should be considered and investigated. Therefore, based on the concept of “neijuan” in the network context, this study redefines involution in a more realistic context, then explores the relationship between involution and the symptoms of anxiety among Chinese college students, with the aim of providing empirical support and a theoretical basis for further studies.

### 1.1. Involution

The term “involution” was introduced by an American anthropologist, Alexander Goldenweiser, to describe a certain plight in agricultural development, in which population growth does not necessarily lead to increases in productivity [4]. In 1965, Clifford Geertz applied this concept to make generalizations about Indonesian agriculture, where despite a limited land area, the labor force continued to grow [5]. In 2000, Huang Zongzhe, a sociologist, used the term “involution” to describe the rural economy in China in the context of a growing population and a decrease in land resources, with daily incomes gradually reducing while the labor force was actually increasing [6]. Since then, the concept of “involution” has been widely used in other academic fields, including economics and politics. In the context of Chinese network, involution usually refers to irrational internal competition or involuntary competition for high-quality resources [7]. For example, if a word requirement for students’ homework is 5000 words, some of the students may choose to write 8000 or 10,000 words, or even more, to obtain better scores and to be more competitive in their future careers. This forces other students to write more words to compete and, finally, everyone works in excess of the teacher’s requirements. However, if the quota for excellence is fixed, many students may work hard to complete their homework but not obtain their expected scores.

Some scholars argue that involution is a social phenomenon in which more people compete for limited resources, resulting in intense and irrational competition [3]. In 2021, Lin stated that involution can be understood as the continuous refinement and complexity of college students’ input in striving for high-quality resources under the conditions of limited college resources, thus diminishing marginal benefits via irrational competition [8].

According to classic social comparison theory (SCT), under intense peer competition, individuals often achieve their goals by comparing themselves with their competitors and outperforming them [9]. Studies show that individuals in the Chinese cultural setting tend to compare themselves with outstanding people around them, a process that is known as upper social comparison, producing a sense of being threatened by social comparisons [10]. The individuals then experience pressure when they feel that sense of being threatened. Some scholars found, in investigations and research on college students, that most students’ pressure comes from social competition, especially in the areas of employment and study [11]. Such findings have shown that when an individual approaches or exceeds the performance of his or her competitors, his or her determination to win is enhanced, and his or her focus shifts from the realization of goals to extra effort. Therefore, greater competitive pressure forces individuals to exercise extra effort in achieving desired goals. The higher the performance and effort of competitors, the more intense the competition.

Achievement motivation refers to the motivation to engage in activities that are significant and challenging, whereby people may achieve perfect and excellent results and surpass others [12]. In 1963, Atkinson put forward the “expectation-value” theory, asserting that people have two psychological tendencies in competition: one tendency is to pursue success and the other tendency is to avoid failure. Previous studies showed that the tendency to pursue success is significantly and positively correlated with the total scores of competition, competitive tendencies, competitive motivations, and competitive contents [13]. There may also be a positive correlation between achievement motivation and involution, as involution is a type of irrational competition.

According to cognitive evaluation theory (CET), external material rewards produce negative reward effects, thereby reducing the individual’s intrinsic motivation; at that point, the individual’s motivation turns to extrinsic motivation [14]. While engaging in involution behavior, individuals who are pursuing potential external material rewards (such as award recognition, excellent evaluations, high scores, etc.), rather than seeking enjoyment or competence or autonomy, are actively involved in exercising extra effort.

Therefore, based on classical social comparison theory, achievement motivation theory, and cognitive evaluation theory, this study proposes three potential types of college students’ involution behaviors—passive involution, achievement-motivated involution, and reward-oriented involution. Passive involution means that people are forced to participate in irrational competition because they do not want to be surpassed by their peers; achievement-motivated involution means that people avoid falling behind their peers in order to improve their competitiveness, and actively participate in irrational competition; reward-oriented involution means that people join in irrational competition in order to obtain more resources and to avoid being overtaken by their peers. These three types of behavior provide categories of involution. According to these three dimensions of involution, we define involution as types of behaviors wherein individuals are under peer pressure or extrinsic motivation to be involved in active or submissive irrational competition for limited social resources. 

We can think of involution as a special kind of competition. Competition is the desire to outperform others in order to enhance reputation [15], or to seek to achieve a dominant position over others in various fields [16]. However, competition is generally active, and its core essence is to gain what a person desires. Involution is active or passive; however, its core essence is force due to peer pressure. Even though active involution (achievement-motivated involution or reward-oriented involution) seems to lead to certain accomplishments, the essential purpose of positive involution is to obtain as many resources as possible so that the individual does not fall behind his or her peers.

### 1.2. Anxiety

The American Psychiatric Association has indicated that anxiety is an individual’s anticipation of an imminent threat, accompanied by changes in stress and physiological conditions [17]. Zhao and Ou believed that anxiety is an emotional state of fear and nervousness that occurs when individuals feel that they cannot achieve their goals or overcome the obstacles they encounter [18]. Some researchers have proposed that anxiety is excessive worry about the unknown, which is a type of emotional disorder. In severe cases, some individuals will experience extreme fear [19].

In this study, anxiety is defined as a comprehensive emotional state of tension, worry, and fear, accompanied by a series of symptoms. such as sweating, tremors, and/or chest tightness during conflicts, setbacks, or unfamiliar situations.

### 1.3. Relationship between Involution and Anxiety

Many scholars have observed that involution or irrational competition may lead to anxiety [10,20]. Involution complicates students’ interpersonal communications, generates mutual hostility, and increases the breaking of rules and moral ethics [10]. One study showed that excessive competition can easily lead to psychological problems, such as depression, anxiety, and interpersonal deterioration [20]. Wu and Zhang pointed out that a “comparative pressure” effect existed among peer groups—the better the peer performs, the greater the perceived pressure [21]. College students were prone to mental health problems, such as depression, anxiety, and stress influenced by intense competition [22]. Some scholars found that if competition was more intense and cooperation was absent, students were hindered to a certain extent in establishing good interpersonal relationships. Meanwhile, peer competition has a negative impact on adolescents’ interpersonal adaptation [23]. Lin pointed out that the involution behavior of college students causes anxiety in terms of study, life, and social interaction, causing great harm to their physical and mental health [10].

Despite the growing evidence for a positive relationship between involution behavior and anxiety symptoms among college students, no previous studies have considered the three different type of involution behavior and their potential relationships with anxiety. In this study, we hypothesize positive relationships between the different types of involutions and anxiety.

### 1.4. The Effect of Demographical Variables on Involution

No previous study has considered the effect of demographic variables on involution. By definition, involution is mainly based on irrational internal competitive behavior for limited resources. Empirically, people from lower social classes have fewer social resources. In this study, several demographic variables related to social status were designed, including family region, college level, and educational background of parents. We assumed that college students whose families live in rural areas, or who attend lower-level universities, or whose parents received less education have fewer resources and are engaged in more involution behaviors. Liu and Liu found that college students of different genders with different majors had different competitive attitudes [24]. Males, and science students generally, tended to hold more benign competitive attitudes. We hypothesized that significant differences may be found in the involution involvement of college students of different genders with different family regions, college levels, academic backgrounds, and parental educational backgrounds.

## 2. Materials and Methods

### 2.1. Participants

Data for this study came from 936 full-time college student volunteers, including undergraduate and graduate students from all parts of China. Three batches of data were collected, the first two of which were mainly used in the preparation and validation of the questionnaires (*n* = 120, 119 valid; *n* = 236, 224 valid). The third batch of data (*n* = 578, 535 valid) was used to explore the relationship between involution and other variables (such as certain demographic variables and anxiety). We excluded those students who responded too quickly and those who failed the polygraph. Becausde this study was mainly based on the third batch of participants, we focused on their characteristics. Table 1 shows the characteristics of the third batch of participants who were included. Because the sample of doctors was too small (*n* = 3) for a formal study, we merged that sample and the master sample into a sample of graduate students in the data processing.

All survey subjects participated voluntarily after signing an informed consent form. This study was approved by the Psychology School of the South China Normal University’s Ethical Review Committee (no. SCNU-PSY-2022-043).

### 2.2. Measures

#### 2.2.1. Involution Involvement

Involution behavior was assessed by a theory-based scale with satisfactory validity and reliability. The scale was developed according to the study’s theory. Two preliminary studies were conducted to assess the validity of the scale. The first preliminary study with 120 samples (119 of which were valid) adjusted the theoretical dimensions and items after item analysis and exploratory factor analysis. The second preliminary study applied the adjusted scale to 236 collected samples (224 of which were valid) to retest the item analysis and the exploratory factor analysis. Items with better factor loading were selected to comprise the final involution behavior scale, which reported satisfactory reliability (α = 0.925) for formal study. The self-report agreement on the 26 statements of the 13 scenarios of daily behavior was included in the scale. To distinguish positive and passive involution behavior, all 13 daily behavior scenarios were presented in both positive and passive expression to complete the 26 items. Example items included the following: “To complete the paper assignment with far more words than required” and “I worry about falling behind others when I see my classmate’s paper assignments exceed the word count requirement” (all items are provided in Appendix A). The involution behavior scale incorporated three dimensions: factor 1 contains 8-item achievement-motivated involution, factor 2 contains 5-item reward-oriented involution, and factor 3 contains 13-item passive involution. See Table 2 for the correlation between each item, each factor, and total score. A formal study with 535 samples was conducted to confirm the validity of the scale via confirmatory factor analysis, as well as to verify the structure of the three factors. As shown in Table 3, all three dimensions were moderately associated with each other (r = 0.26–0.69, *p* < 0.01), and they were highly related to the whole scale (r = 0.67–0.87, *p* < 0.01), indicating a validated structure. Confirmatory factor analysis was used to verify the three factor model of involution. The KMO coefficient was 0.936 and the Bartlett Spherical Test result was significant (*p* < 0.001), indicating that is the model was suitable for factor analysis. The three-factor model of the involution scale is shown in Figure 1. The acceptable fix indices (see Table 4) and the high factor loadings confirmed that the scale is well structured. Responses to these items were collected on a scale anchored by 1 = “disagree strongly” and 5 = “agree strongly”, and they were totaled for a score ranging from 26 to 130, with higher scores indicating more involvement of involution behavior.

#### 2.2.2. Self-Rating Anxiety Scale (SAS)

We measured the participants’ anxiety symptoms with the 20-item Self-Rating Anxiety Scale (SAS) that was developed by Zung and validated by other scholars, in the Chinese version SAS [25,26,27]. All items were ranked on a four-point Likert scale, ranging from 1 = “occasionally” to 4 = “always”. Negative 5 items were reverse-coded, and ratings on each item were totaled to establish a score ranging from 20 to 80 (Cronbach’s α is 0.93), with higher scores indicating more anxiety symptoms. The norm of Chinese version SAS is that scores of 50 to59 scores indicate mild anxiety; scores of 60 to 69 indicate moderate anxiety; and scores over 70 indicate severe anxiety.

### 2.3. Statistical Analyses

Descriptive statistics, difference tests (independent sample *t*-test and two-way ANOVA), correlation analysis, and regression analysis were used in this study. Descriptive statistics were performed to illustrate the overall reflection of involution involvement and anxiety among the college students. Then, difference testing was implemented to analyze the difference of involution involvement in various demographic variables. Finally, correlation analysis and regression analysis were applied to explore the relationship between involution and anxiety. All data analyses were conducted in SPSS 20.0 (IBM Corp., Armonk, NY, USA) and Mplus 8.3 (Muthén & Muthén, Los Angeles, CA, USA) [28].

## 3. Results

### 3.1. Descriptive Statistics

Table 5 shows the descriptive statistics of involution involvement and anxiety. The average scores of active involution, especially achievement-motivated involution, were much higher than those of passive involution, which indicated that achievement-motivated involution was the greatest contributor to students’ average involution involvement, while passive involution contributed the least. The high average score and the large standard deviation showed that the college students were generally heavily engaged in frequent involution, although there were large individual differences.

The average anxiety score was 50.84, indicating that most of the college students were suffering from anxiety. Specifically, 50.1% of the college students had an anxiety score of fewer than 50 points; 23.4% had a score of 50 to 59 points, indicating a mild anxiety state; 17.3% had a score of 60 to 69 points, indicating a moderate anxiety state; 9.2% had a score of more than 69 points, indicating a severe anxiety state.

### 3.2. Demographic Differences among College Students

Table 6 shows the demographic differences in involution involvement among college students. Only the college level factor was found to be a significant difference in the students’ involution involvement.

Table 7 shows the results of post hoc multiple comparisons by college level (according to the college entrance examination score line to distinguish the college level: Project 985 ≥ Project 211 ≥ The ordinary first batch of universities ≥ The second batch of universities ≥ junior colleges). Students from Project 985 were involved in significantly less involution behavior than were students from lower-level schools (*p* < 0.01). Similarly, students from Project 211 were involved in significantly less involution behavior than were those from lower-level schools (*p* < 0.01). In addition, students from the second batch of universities were involved in more involution behavior than were junior colleges students.

Taking gender and family region (urban areas, township areas, and rural areas) as independent variables, and reward-oriented involution as the dependent variable, a 2 × 3 two-way analysis of variance was conducted to explore the main effects and the interaction effects of each independent variable. The results are shown in Table 8.

The main effects of gender and family region were not significant and the interaction effect was significant. A simple effect analysis was conducted, as shown in Figure 2. Males whose family regions were located in urban areas were more involuted by external rewards than were females; females whose family regions were townships and rural areas were more involuted by external rewards than were males.

### 3.3. Differences in Gender and Major Types of College Students’ Involution Induced by External Rewards

Taking gender and academic background (social science, humanities, engineering, natural science, and medicine) as independent variables, and the reward-oriented involution as the dependent variable, a 2 × 5 two-way ANOVA was performed to explore the main effects and the interaction effects of each independent variable. The results are shown in Table 9.

The main effects of gender and academic background were not significant, but the interaction effect was significant. A simple effect analysis was conducted and the results are shown in Figure 3. For social sciences, humanities, natural sciences, and medical students, males were involved in higher levels of reward-oriented involution behavior than were females; for engineering students, females were involved in higher levels of reward-oriented involution behavior than were males.

### 3.4. Correlation Analysis between Involution and Anxiety of College Students

The two-tailed test of Pearson product-moment correlation was implemented to analyze the pairwise correlation of factors. Table 10 showed that the students’ overall involution involvement was mildly and positively correlated with anxiety (r = 0.30, *p* < 0.001); passive involution was moderately correlated with anxiety (r = 0.40, *p* < 0.001); reward-oriented involution was weakly correlated with anxiety (r = 0.09, *p* < 0.001); and the correlation between achievement-motivated involution and anxiety was not significant (r = −0.009, *p* > 0.05).

### 3.5. Regression Analysis of College Students’ Involution and Anxiety

To further investigate the relationship between involution involvement and anxiety symptoms among the college students, multiple regression was applied. The regression analysis was carried out with passive involution, achievement-motivated involution, and reward-oriented involution as independent variables, and anxiety as the dependent variable. As shown in Table 11, the regression equation was significant. Passive involution, achievement-motivated involution, and reward-oriented involution together accounted for 17% of the students’ anxiety symptoms. Passive involution caused a significant positive predictive effect on anxiety, while achievement-motivated involution caused a significant negative predictive effect on anxiety. However, reward-oriented involution was not significantly predictive of anxiety symptoms among the college students.

## 4. Discussion

The results of this study show that, overall, the college students scored highest on achievement-motivated involution, followed by reward-oriented involution and passive involution. The same trend was in evidence for both males and females. The college students scored higher in the two dimensions of active involution, indicating that their current involution behavior was mainly intended as “active attack” (taking the initiative to participate in academic and social competition with peers). The high average score for achievement-motivated involution suggested tha the college students were mainly driven and inspired by success and accomplishment, and to be involved in particular competitions to attain certain goals. With the popularization of higher education, social competition is becoming more and more pronounced, in that higher requirements are imposed on college graduates when they are being interviewed. Therefore, with limited social resources to achieve their goals, college students are actively involved in competition to gain advantages in future studies and work, by surpassing their competitors. Second, some college students are involved in competition to obtain external rewards, such as receiving extra credits, rather than to enhance their competitiveness. Finally, college students engage least in passive involution behavior, indicating that they express more positive determination and take the initiative in improving themselves, rather than simply following trends.

### 4.1. The Relation of College Level and Involution Involvement

The demographic differences in involution involvement showed that students’ college levels significantly affected the involvement of their involution behavior.

Specifically, students in Project 985 and Project 211 had the lowest degree of involution, which could be due to internal motivation. Some past studies have shown that students with stronger internal motivation have better learning performances than those with stronger extrinsic motivation [29,30]. Better academic performance means being able to be accepted into better colleges. Students in high level colleges have higher internal motivation. Compared with students with lower college levels, high level students’ participation in competition is mainly derived from internal motivation to learn and to obtain more knowledge to enrich their experiences and insights, and thus to establish themselves as more adaptive people. However, the competition that results from internal motivation is not within the scope of involution, according to the definition.

In addition, high-level universities are endowed with greater educational resources [31], such as more advanced experimental equipment, more scholarships, and more admission quotas for further study without examination. Therefore, students at such universities receive greater academic, material, and social resources without competition, and their involution involvement is lower than that of students from other colleges.

The first and second batches of ordinary undergraduates reported the highest involution involvement among all students. Driven by the pressures of upward comparisons, they are forced to compete in the hope of surpassing peers, via their efforts to obtain limited resources or attain greater senses of accomplishment. Meanwhile, compared with the students from higher level schools, such students need to surpass more peers to gain access to limited resources, leading to more competition and a higher involvement of involution.

The third batch of students, comprising of undergraduate and junior college students, demonstrated a medium level of involution involvement, implying that their achievement motivation and initiative are relatively lower than those of students from other colleges. When encountering competition, they are preferred to “lie down” persons (or, in Chinese, “tang ping”, a recent popular phrase that describes a state of enduring being surpassed by others); thus, they are engaged in relatively less involution. On the other hand, the educational and social resources for such students are more limited, so they are face pressure when competing with peers from better educational backgrounds. Some of these students will still join in competition and engage in involution.

### 4.2. The Relation of Passive Involution, Reward-Oriented Involution, Active Involution-Achievement-Motivation and Anxiety

This paper has explored the relationship between college students’ involution involvement and anxiety symptoms, and found that passive involution, reward-oriented involution, and total involution are significantly and positively correlated with anxiety symptoms. Further analysis showed that college students’ passive involution is positively predictive of their anxiety symptoms, while active achievement-motivation involution behavior is negatively predictive of their anxiety symptoms. These results suggest that higher passive involution involvement is associated with more anxiety symptoms, which corresponds with the previous suggestions about high school students, higher vocational students, and college students; i.e., that passive involution, as individuals are forced to participate in involution due to peer pressure, is more likely to result in mental problems, such as anxiety [10,20,32]. This connection can be explained by Festinger’s classic social comparison theory. In situations of fierce peer competition, individuals often achieve their goals by comparing themselves with their competitors and surpassing the achievements of their competitors [11]. When an individual is compared with the outstanding people around him or her, a threat of social comparison occurs, which leads to excessive competitive behavior and endangerment of students’ mental health [12]. When college students are motivated to improve in competitiveness or to achieve goals, competition generates less anxiety. The results of this study confirmed the view that is stated in Deci’s cognitive evaluation theory; that is, involution activated by intrinsic motivation has positive effects for students’ mental health, while involution stimulated by external material rewards has negative effects [16].

### 4.3. Contributions

Although “involution” is widely used in daily life, its academic definition is divorced from a real context. The lack of measurement scales disadvantages scholars in carrying out quantitative research and in-depth discussions. In response to this problem, this study’s innovations are as follows:

First, this study creatively defined “involution” as the irrational and internal competitive behavior of competitors due to external pressure or external motivation under circumstances of limited social resources. The definition incorporates both academic definitions and a realistic context.

Second, this study determined the theoretical dimension of involution through a literature analysis and interview investigations, then compiled an involution behavior scale for college students that was suitable for the network cultural background. The successful development of this scale provided a good quantitative tool for future research.

Finally, academic research on involution was extremely scarce, although its pros and cons have been debated on the internet. This study provided practical evidence that involution was not completely harmful; its potential impact on anxiety depended on different motivations.

### 4.4. Implications

This study has several implications. First, involution is not always detrimental; the consequences of involution depend on how people perceive it. In the increasingly competitiveness of modern society, people may not view involution as a negative. Rather, involution is not completely harmful in a fiercely competitive environment in which there is no way out. In such circumstances, it is better to adjust mentality and motivation, and to view competition as a means of improvement rather than as a cause of feeling helpless or engaging in behavior for purely external rewards. This perspective may enable people to learn how to live in harmony with involution and not to be overly anxious.

### 4.5. Limitations

We hope that future research on involution can pursue this paper’s directions in-depth. Firsy, the target group of this study was Chinese college students, so all aspects of the study were specially designed. However, the phenomenon of involution does not take place only in groups of college students. Future research can expand the groups of subjects and consider involution behaviors in other areas, incorporating new scenarios to further enrich the study of involution.

Second, this study showed the effect of involution on anxiety, but the psychological mechanisms that affect involution have not been studied. Future researchers may incorporate additional psychological variables (such as cognitive style, personality, attribution, psychological resilience) in their works on involution behaviors.

## 5. Conclusions

This study aimed to explore the internal structure of involution, which demographic variables affect involution, and its relationship with anxiety. The results were as follows: (1) The involution behavior of the Chinese college students could be divided into three types: the passive involution, reward-oriented involution, and achievement-motivated involution; (2) Significant differences in the involvement of involution existed at the college level; (3) Three motivations that resulted in involution, from primary to secondary, were achievement-motivation, reward-orientation, and passive engagement; and (4) Passive involution, reward-oriented involution, and the total scores for the involution behavior of the college students were significantly and positively correlated with anxiety. Among the three types of involution behavior, the college students’ passive involution had a significant and positive predictive effect on their anxiety, while achievement-motivated involution had a significant and negative predictive effect.

## Figures and Tables

**Figure 1 ijerph-19-09826-f001:**
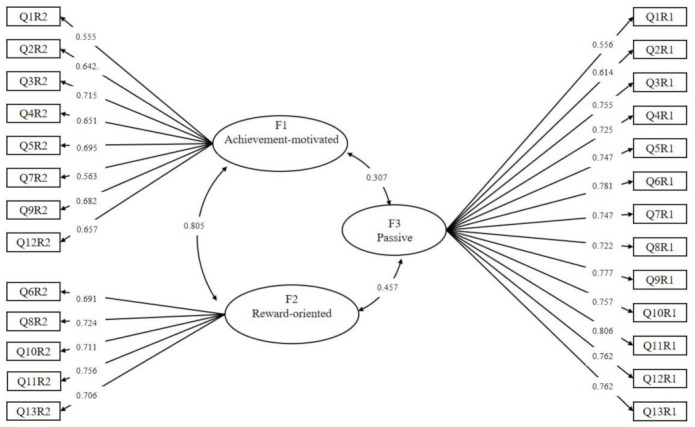
The three-factors model of the college students’ involution.

**Figure 2 ijerph-19-09826-f002:**
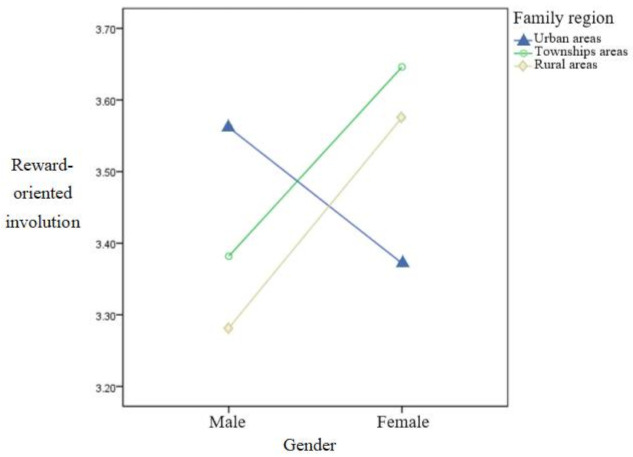
The interaction diagram of gender and family region for reward-oriented involution. *n* = 535.

**Figure 3 ijerph-19-09826-f003:**
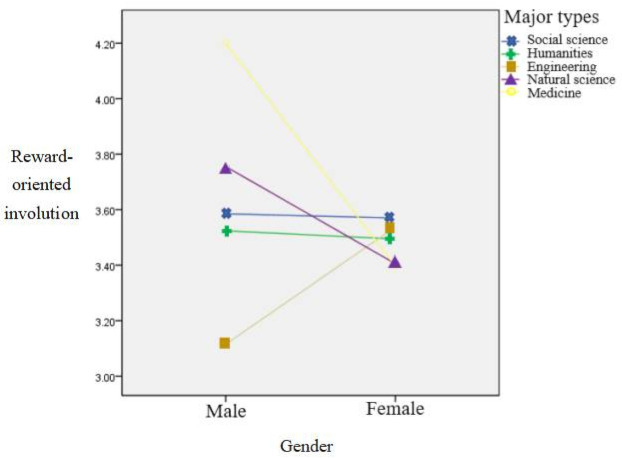
Interaction diagram of gender and academic background for reward-oriented involution. *n* = 535.

**Table 1 ijerph-19-09826-t001:** Characteristics of the study sample.

Variable	Characteristics		Sample of Formal Study	
Gender	Male		150	
	Female		385	
Academic degree	Undergraduate	Grade 1	72	443
		Grade 2	111	
		Grade 3	139	
		Grade 4	89	
	Master	Grade 1	21	89
		Grade 2	49	
		Grade 3	19	
	Doctor	Grade 4	3	
Family region	Urban		218	
	Township area		137	
	Rural area		180	
College level	Project 985		39	
	Project 211		59	
	The ordinary first batch		65	
	The second batch		138	
	The third batch		27	
	Junior college		207	

Note. *n* = 535.

**Table 2 ijerph-19-09826-t002:** Correlations of items and total score.

	Factor	F1	F2	F3	Total
Item					
Q1R2		0.63 **			0.51 **
Q2R2		0.71 **			0.53 **
Q3R2		0.75 **			0.46 **
Q4R2		0.70 **			0.40 **
Q5R2		0.73 **			0.49 **
Q7R2		0.64 **			0.44 **
Q9R2		0.73 **			0.42 **
Q12R2		0.70 **			0.48 **
Q6R2			0.76 **		0.59 **
Q8R2			0.80 **		0.58 **
Q10R2			0.79 **		0.58 **
Q11R2			0.80 **		0.62 **
Q13R2			0.76 **		0.55 **
Q1R1				0.61 **	0.54 **
Q2R1				0.66 **	0.62 **
Q3R1				0.78 **	0.69 **
Q4R1				0.76 **	0.65 **
Q5R1				0.77 **	0.67 **
Q6R1				0.80 **	0.71 **
Q7R1				0.77 **	0.67 **
Q8R1				0.75 **	0.64 **
Q9R1				0.79 **	0.68 **
Q10R1				0.77 **	0.66 **
Q11R1				0.81 **	0.69 **
Q12R1				0.78 **	0.66 **
Q13R1				0.75 **	0.66 **
Crombach’s α		0.845	0.840	0.937	0.925

Note. *n* = 535; ** *p* < 0.01.

**Table 3 ijerph-19-09826-t003:** Internal consistency validity of the involution measurement.

	Passive Involution	Achievement-Motivated Involution	Reward-Oriented Involution	Total
Passive involution	1			
Achievement-motivated involution	0.263 **	1		
Reward-oriented involution	0.395 **	0.691 **	1	
Total	0.870 **	0.671 **	0.747 **	1

Note. *n* = 535; ** *p* < 0.01.

**Table 4 ijerph-19-09826-t004:** Confirmatory Factor Analysis Model Fit Indices.

χ^2^	df	χ^2^/df	*p*	RMSEA	SRMR	CFI	TLI
1079.233	296	3.646	<0.001	0.070	0.054	0.894	0.884

Note. *n* = 535; RMSEA = root mean square error of approximation; SRMR = standardized root mean square residual; CFI = comparative fix index; TLI = Tucker-Lewis index.

**Table 5 ijerph-19-09826-t005:** Descriptive statistics among study variables.

	*n*	Min	Max	M	SD
Passive involution	535	1.00	5.00	2.96	0.92
Achievement-motivated involution	535	1.00	5.00	3.85	0.67
Reward-oriented involution	535	1.00	5.00	3.49	0.91
Involution	535	26	130	86.74	17.37
Anxiety	535	25.00	82.50	50.84	12.78

Note. *n* = 535.

**Table 6 ijerph-19-09826-t006:** Difference test of involution involvement in various demographic variables.

	SS	df	MS	F(t)
Gender		1		0.75
College level	7074.92	5	1414.98	4.86 ***
Family region	306.61	2	153.30	0.51
Academic background	2021.22	4	505.30	1.68
Parental education background	2672.24	9	296.92	0.98
Family income	639.44	4	159.86	0.53
Subjective social class	2621.68	4	655.42	2.19

Note. *n* = 535; SS = sum of squares; MS = mean squares; *** *p* < 0.001.

**Table 7 ijerph-19-09826-t007:** Post hoc multiple comparisons of involution involvement at the college level.

I	J	(I) Mean	(J) Mean	Mean Difference (I−J)	Critical Value
Project 985	Project 211	81.69	79.80	1.89	0.591
Project 985	The ordinary first batch of universities	81.69	89.25	−7.56	0.029 *
Project 985	The second batch of universities	81.69	91.01	−9.32	0.003 **
Project 985	The third batch of universities	81.69	88.81	−7.12	0.096
Project 985	Junior colleges	81.69	85.76	−4.07	0.173
Project 211	The ordinary first batch of universities	79.80	89.25	−9.45	0.002 **
Project 211	The second batch of universities	79.80	91.01	−11.21	0.000 ***
Project 211	The third batch of universities	79.80	88.81	−9.01	0.023 *
Project 211	Junior colleges	79.80	85.76	−5.96	0.018 *
The ordinary first batch of universities	The second batch of universities	89.25	91.01	−1.76	0.493
The ordinary first batch of universities	The third batch of universities	89.25	88.81	0.44	0.912
The ordinary first batch of universities	Junior colleges	89.25	85.76	3.49	0.151
The second batch of universities	The third batch of universities	91.01	88.81	2.2	0.542
The second batch of universities	Junior colleges	91.01	85.76	5.25	0.005 **
The third batch of universities	Junior colleges	88.81	85.76	3.05	0.382

Note. *n* = 535; * *p* < 0.05, ** *p* < 0.01, *** *p* < 0.001.

**Table 8 ijerph-19-09826-t008:** Two-way ANOVA results of reward-oriented involution involvement with respect to gender and family region.

	SS	df	MS	F
Gender	1.52	1	1.52	1.85
Family region	0.43	2	0.22	0.26
Gender × Family region	5.89	2	2.94	3.59 *

Note. *n* = 535; * *p* < 0.05.

**Table 9 ijerph-19-09826-t009:** Two-way ANOVA results of reward-oriented involution in terms of gender and academic background.

	SS	df	MS	F
Gender	1.41	1	1.41	1.73
Major types	6.21	4	1.55	1.91
Gender × Major types	9.20	4	2.30	2.83 *

Note. *n* = 535; * *p* < 0.05.

**Table 10 ijerph-19-09826-t010:** Correlations among study variables.

Factor	Passive Involution	Achievement-Motivated Involution	Reward-Oriented Involution	Total Involution	Anxiety
Passive involution	1				
Achievement-motivated involution	0.26 ***	1			
Reward-oriented involution	0.40 ***	0.69 ***	1		
Total involution	0.87 ***	0.67 ***	0.75 ***	1	
Anxiety	0.40 ***	−0.009	0.09 ***	0.30 ***	1

Note. *n* = 535; *** *p* < 0.001.

**Table 11 ijerph-19-09826-t011:** Regression analysis of anxiety on involution involvement (*n* = 535).

	Anxiety		
	β	R^2^	F
Passive involution	0.43 ***	0.17	37.43 ***
Achievement-motivated involution	−0.13 *		
Reward-oriented involution	−0.08		

Note. *n* = 535; * *p* < 0.05, *** *p* < 0.001.

## Data Availability

The data are not publicly available, due to privacy restrictions.

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
