# Peer review of "Does Involution Cause Anxiety? An Empirical Study from Chinese Universities"

_ijerph, 2022, doi:10.3390/ijerph19169826_

Round 1
Reviewer 1 Report
The authors investigated the association of involution with anxiety amongst Chinese college and university students. Some developed and validated questionnaires were used in conjunction with various demographic data of the students. It was reported from the findings of the study that involution behaviour of college students can be divided into three types: passive involution, reward-oriented involution and motivated-achievement involution, significant involution involvement differences existed at the college level, and the passive involution, reward-oriented involution, and total scores of involutions of college students were significantly positively correlated with anxiety among other findings. Overall, it is an interesting research and I congratulate the authors for their effort in carrying out this investigation. There are a few issues that need to be addressed to improve the quality of the manuscript. Below are my suggestions:
1. The abstract should highlight the main issue leading to the conduct of the study as well as the implications of the findings of the study to the stakeholders.
2. L36-Fix the citation problem
3. L72-74- Involvement or involution?
4. L156- Remove ‘been’
5. L185- Validate replace with ‘validated’
6. L186-2 replace with two
7. L208-209- Strongly replace with strong
8. L351-363. Citations?
Author Response
Dear Reviewer 1:
Thanks for providing us with this great opportunity to submit a revised version of our manuscript. We appreciate the detailed and constructive comments provided by the referees. We have carefully revised the manuscript by incorporating all the suggestions by the review panel. Here are the details of our changes:
Reply to Reviewer 1:
Comment 1. Moderate English changes required.
Response 1. Thank you for the detailed review. We have carefully revised the grammar and style of the manuscript by an English-major graduate student.
Comment 2. The abstract should highlight the main issue leading to the conduct of the study as well as the implications of the findings of the study to the stakeholders.
Response 2. L15-20- We have added to the abstract why we conducted the study and the implications of the findings of the studyto the stakeholders.
Comment 3. L36-Fix the citation problem.
Response 3. L41- We've fixed that citation problem.
Comment 4. L72-74- Involvement or involution?
Response 4. L77-79- I'm sorry this is our mistake, it should be involution here, we have made corrections.
Comment 5. L156- Remove ‘been’.
Response 5. L-175-Thank you very much for your attention, we have removed the ‘been’.
Comment 6. L185- Validate replace with ‘validated’.
Response 6. L219- Thanks a lot for your tips, we have replaced Validate with 'validated'.
Comment 7. L186- 2 replace with two.
Response 7. L221- Thanks a lot for your suggestion, we have replaced 2 with Two.
Comment 8. L208-209- Strongly replace with strong.
Response 8. L249-250- Thank you very much for reminding, considering that the text should be consistent with Appendix A, we replace ‘strongly disagreement’with ‘disagree strongly’and ‘strongly disagreement’ with ‘agree strongly’.
Comment 9. L351-363. Citations?
Response 9. L399-414- We add citations, mainly on the relationship between internalmotivation and academic performance (because good academic performance means admission to higher-level universities, in order to prove that students in higher-level schools have higher internalmotivation), and to demonstrate that for a Chinese university, the higher the level is, the more resources it has.
The above is our response and modification to the reviewer 1. We hope this revised manuscript has addressed your concerns, and look forward to hearing from you.
Sincerely,
The Authors
Reviewer 2 Report
The paper shows interesting results in line with the objectives set.
In principle, the analyses seem to be the most appropriate, although below I indicate some aspects that should be added.
About theory and the conclusions drawn from this research:
In the conclusions, it is noted that passive involution, reward-oriented involution, and total involution are significantly positively correlated with anxious symptoms.
Whereas active involution-achievement motivation is negatively predictive of their anxious symptoms.
The concept of involution is widely explained, but it is necessary to relate it at least theoretically in order to have a clearer definition with other constructs that are better known at an international level, such as emotional intelligence or resilience. This would improve the understanding of the value of these constructs when generalising them to other populations and contexts.
About the empirical study:
When describing the selection of the sample, it should be specified that the subjects are volunteers.
The composition of the sample is unbalanced in terms of the number of undergraduates (443) versus graduates (89), where 3 doctoral students are also included.
Are the subjects participating in the first analyses included again in the subsequent analyses (sample of 532 subjects)?
It is not clear that the terms "pretest" and "posttest" are the most appropriate for this occasion. No treatment or intervention is applied between the two occasions.
I consider it more appropriate to use the term pilot study, or preliminary studies. This is a process of designing and validating a questionnaire.
The Kaiser-Meyer-Olkin measure of sampling adequacy (KMO coefficient) is missing in the factor analysis, and it would be good to provide it.
In the calculation of the Crombach's internal consistency coefficient (Reliability) it would be good to have a table showing the Homogeneity Indexes of the items. In this way, it can be observed how their conservation or elimination favours the internal consistency of the questionnaire.
One last suggestion. In Appendix A (Table A1) I have noticed that for each situation, the negative or passive items are included in the same space as those expressing an active or positive attitude. In order that this may not produce some bias in the responses, my suggestion would be that, for future occasions, they should be separated. So that items referring to different situations are interspersed.
For example, item 1 could be stated:
I choose some courses with high grades but no interest because I’m worried that my GPA would fall be-hind others when I see my classmates taking these courses
Then continue with other items, and later introduce:
I choose some courses with high grades but no interest because I actively choose these courses to get a higher GPA
I hope you find these suggestions and indications useful. The topic is interesting and it would be desirable that it be studied on more occasions and in more countries.
Author Response
Dear editors and reviewer 2:
Thanks for providing us with this great opportunity to submit a revised version of our manuscript. We appreciate the detailed and constructive comments provided by the reviewer 2. We have carefully revised the manuscript by incorporating all the suggestions by the review panel. Here are the details of our changes:
Reply to Reviewer 2:
Comment 1. In the conclusions, it is noted that passive involution, reward-oriented involution, and total involution are significantly positively correlated with anxious symptoms. Whereas active involution-achievement motivation is negatively predictive of their anxious symptoms. The concept of involution is widely explained, but it is necessary to relate it at least theoretically in order to have a clearer definition with other constructs that are better known at an international level, such as emotional intelligence or resilience. This would improve the understanding of the value of these constructs when generalising them to other populations and contexts.
Response 1. L121-141- We associate passive involution with classical social comparison theory, motivated-achievement involution with achievement motivation theory, reward-oriented involution with cognitive evaluation theory, and jointly explain the three dimensions of involution, and finally involution with competition for comparison, the definition of involution is explained to facilitate people's understanding of involution.
Comment 2. When describing the selection of the sample, it should be specified that the subjects are volunteers.
Response 2. L194- Thank you very much for your suggestion, we have specified that the subjects are volunteers.
Comment 3. The composition of the sample is unbalanced in terms of the number of undergraduates (443) versus graduates (89), where 3 doctoral students are also included.
Response 3. L203-205- Thanks for your question. Since the sample of doctoral students is too small (n=3) in formal study, we merge it and sample of master into the sample of graduate students in the later data processing. New explanations have been added to our manuscript.
Comment 4. Are the subjects participating in the first analyses included again in the subsequent analyses (sample of 532 subjects)?
Response 4. L197-201- A total of three batches of data were collected in this study, of which the first two batches of data were mainly used for the preparation and validation of the questionnaires (n1= 120, 119 valid; n2 = 236, 224 valid), and the third batch of data (n3 = 578, 535 valid) was used to explore the relationship between involution and other variables (such as some demographic variables and anxiety). Therefore, only the third batch of data is actually used in the formal study.
Comment 5. It is not clear that the terms "pretest" and "posttest" are the most appropriate for this occasion. No treatment or intervention is applied between the two occasions.I consider it more appropriate to use the term pilot study, or preliminary studies. This is a process of designing and validating a questionnaire.
Response 5. L221-228- Thank you very much for your suggestion, we have replaced first pre-test with preliminary study 1, second pre-test with preliminary study 2and post test with formal study.
Comment 6. The Kaiser-Meyer-Olkin measure of sampling adequacy (KMO coefficient) is missing in the factor analysis, and it would be good to provide it.
Response 6. L244-246- We have added KMO coefficients and the Bartlett Spherical Test resultto the manuscript.
Comment 7. In the calculation of the Crombach's internal consistency coefficient (Reliability) it would be good to have a table showing the Homogeneity Indexes of the items. In this way, it can be observed how their conservation or elimination favours the internal consistency of the questionnaire.
Response 7. L254- Thank you very much for your professional opinion, and we have added Table 2 of the correlations of items and total score in the manuscript.
Comment 8. One last suggestion. In Appendix A (Table A1) I have noticed that for each situation, the negative or passive items are included in the same space as those expressing an active or positive attitude. In order that this may not produce some bias in the responses, my suggestion would be that, for future occasions, they should be separated. So that items referring to different situations are interspersed.
Response 8. Since we have used Appendix A as a measurement tool in our research, there is no way for us to make changes to Appendix A, but considering your reasonable and scientific advice, we have newly added Appendix B, which is a modification of Appendix A, and suggested that subsequent researchers may consider using Appendix B.
The above is our response and modification to the reviewer 2. We hope this revised manuscript has addressed your concerns, and look forward to hearing from you.
Sincerely,
The Authors